# Research on Tool Wear Based on 3D FEM Simulation for Milling Process

**Zhibo Liu [1]** , **Caixu Yue [1,\*], Xiaochen Li [1], Xianli Liu [1], Steven Y. Liang [2] and Lihui Wang [3]**

[1]  The Lab of National and Local United Engineering for High-Efficiency Cutting & Tools, Harbin University of Science and Technology, Harbin 150080, China; simplelzb@163.com (Z.L.); 18845156457@163.com (X.L.); xlliu@hrbust.edu.cn (X.L.)

[2]  George W. Woodruff School of Mechanical Engineering, Georgia Institute of Technology, Atlanta, GA 30332, USA; steven.liang@me.gatech.edu

[3]  KTH Royal Institute of Technology, 10044 Stockholm, Sweden; lihui.wang@iip.kth.se

\*  Correspondence: yuecaixu@hrbust.edu.cn; Tel.: +86-451-8639-0583

**Abstract:** In the process of metal cutting, the anti-wear performance of the tool determines the life of the tool and affects the surface quality of the workpiece. The finite element simulation method can directly show the tool wear state and morphology, but due to the limitations of the simulation time and complex boundary conditions, it has not been commonly used in tool life prediction. Based on this, a tool wear model was established on the platform of a finite element simulation software for the cutting process of titanium alloy TC4 by end milling. The key technique is to embed different types of tool wear models into the finite element model in combination with the consequent development technology. The effectiveness of the tool wear model was validated by comparing the experimental results with the simulation results. At the same time, in order to quickly predict the tool life, an empirical prediction formula of tool wear was established, and the change course of tool wear under time change was obtained.

**Keywords:** milling; finite element simulation; tool wear; tool life prediction

## 1. Introduction

With the continuous development of the manufacturing industry, the surface performance of parts is more and more demanding. Tool wear directly affects the workpiece processing quality, tool life, and processing cost, and has accurate time-varying characteristics, so many scholars have carried out research on tool wear. Scholars mostly use experimental method or analytic method to predict tool wear, but few adopt the method of finite element simulation to study tool wear, the main reason being that tool wear is a complicated process. Finite element simulation that is carried out from the engineering direction must neglect many factors, which also causes many limitations including a lengthy simulation and complex boundary conditions and so on. With the improvement of computer hardware calculation speed and software simulation efficiency, the finite element simulation method can effectively simulate the course of tool wear by considering the characteristics of milling process such as depth of cut variation.

In the research on tool abrasion simulation in analytic method, scholars find the tool abrasion is influenced by the cutting parameter, geometrical parameter of tool, cutting edge form of tool, cooling method, lubrication method, etc. In order to reveal the influence of cutting parameter on tool abrasion, Choudhury et al. [1] first established the analytic model of tool abrasion for the lathe tool without coating and researched the influence of feed rate and main shaft's rotating speed on tool abrasion. Zhou at al. [2] calculated the stress distribution of tool nose in finite element method

and finally worked out the optimal tool chamfering parameter. With regard to the influence of tool's geometrical shape on tool abrasion, Denkena et al. [3] obtained the service life graph of lathe tools with different cutting edges microstructure in experiment research method, and later Rathod et al. [4] established a prediction model of the abrasion of lathe tool's flank surface by considering the influence of rake angle and clearance angle on abrasion of lathe tool's flank surface. Further, Liu et al. [5] established the geometric model in which the rake angle, clearance angle, and cutting edge radius were comprehensively considered; they defined the evaluation index of tool abrasion loss–tool abrasion volume fraction, and found the abrasion was closely related to rake angle, but unrelated to the radius of cutting edge through experiment. The cutting force and cutting temperature are key factors. Zhang et al. [6], based on the energy consumption method, established the abrasion model of flank surface to research the influence of cutting force on tool abrasion. The contact status between tool cuttings or tool technique also influences tool abrasion. Pervaiz et al. [7], in the cutting process, cooled cutting process with the mixture of air and vegetable oil, and found that tool abrasion speed declined obviously. Moreover, PC Wanigarathne et al. [8] researched the influence of temperature on tool abrasion through experiment and established the coupling relationship between cutting temperature and cutting force in the cutting process.

Regarding the complex cutting process, the finite element simulation method is adopted to determine the tool abrasion shape and abrasion depth intuitively. In the research on the simulation of tool abrasion in the finite element method, most scholars obtain abrasion results by compiling subprograms. Attanasio et al. [9] carried out the finite element simulation analysis to the tool abrasion in processing process through DEFORM software and proposed different abrasion calculation models to account for tool abrasion at different temperatures. Their research results improved the simulation precision. Binder et al. [10], based on the abrasion calculation through DEFORM software, carried out the finite element simulation to the tool with different coating, and expanded the application of finite element in mechanical processing field by considering the element inversion and other factors in the abrasion process. Further, Lotfi et al. [11], based on the secondary development of deform software, proposed a three-dimensional finite element simulation method of tool abrasion with ultrasound-assisted rotary turning, and researched the influence of vibration and rotational motion on tool abrasion and heating. As a result, the surface quality and cutting force improved greatly, compared with that of conventional turning. Faini et al. [12] researched the tool abrasion in drilling proprocess andt the abrasion appearance of bore bit after drilling and the tool abrasion loss after the secondary development of DEFORM software. Lijing et al. [13] utilized the secondary development technology of ABAQUS to research the tool abrasion due to turning processing or Polygon cutting and proposed a prediction method of tool abrasion based on ABAQUS software. Yujing et al. [14] got the tool abrasion loss in two-dimensional milling process through The Third Wave of finite element simulation software, and embedded diffusion model in a finite element simulation to verify the accuracy of model. Malakizadi et al. [15] also proposed a kind of modeling method of three-dimensional tool abrasion finite element. The difference was that the calculation efficiency of abrasion model improved greatly. Such a vale was calculated through an independent MATLAB code, instead of iteration, to shorten simulation time and improve simulation efficiency. Giovanna et al. [16] put forward a new approach to big data for milling cutter wear classification based on signal imaging and deep learning, which can accurately obtain and study the characteristics of the original data, and it can effectively classify wear conditions of a milling cutter. The deep learning theory put forward by Hinton [17] provides a new path to processing and analysis industrial big data, the problem that shallow layer neural network convergence speed is uncontrollable to cause the local optimum was solved. Bi etc. selected cutting depth, cutting width, and feed as experimental parameters that influence tool wear and designed a single factor experiment to construct the prediction model of tool wear with BP neural network algorithm. The experimental results and simulation results have good consistency. Zhang et al. [18] selected the time-domain eigenvalue of the vibration signal of tool wear to realize the prediction of tool wear with the fuzzy C-means clustering algorithm.

To sum up, the current research focuses on predicting tool abrasion in metal cutting by empirical modeling and physics-based modeling. There are few research works predicting tool abrasion course and tool service life in the finite element method, and the analytical method does not display the tool abrasion intuitively. Scholars center on predicting specific abrasion loss, and there are less research studies on the tool abrasion position and the prediction method of wear appearance. Based on the existing cutting finite element simulation and theoretical model of tool abrasion, the abrasion of flank surface on end mill is simulated in finite element method for the milling process of titanium alloy Ti6Al4V. In the meantime, the simulation results are put in empirical formula to predict the service life of tool, and the milling practice is adapted to predict the precision of model.

## 2. Compilation of Subprogram and Establishment of Milling Finite Element Simulation Model

### 2.1. Design of Subprogram for the Tool Abrasion Prediction

The computation program flow of tool abrasion course is shown in Figure 1. In order to accelerate simulation speed and improve simulation efficiency, the traditional iterative algorithm is not adopted. The subprogram of users will become a part of finite element simulation, through which the value of node stress and node temperature is withdrawn from simulation results and calculated to work out the abrasion loss of cutting edge. The wear model takes temperature and stress as inputs. The node coordinates are updated based on the abrasion loss after calculation to update the geometrical shape of tool and reach abrasion status. Later, the abrasion value is put in an empirical formula to calculate the tool abrasion course and complete predicting service life of tool. In the finite element simulation model with three-dimensional cutting, since the tool abrasion loss is far less than the element dimension, the node displacement is adopted to reach the tool abrasion effect, without considering the element deletion of tool cutting edge incurred due to abrasion in this research, in order to improve the simulation speed and adapt to the application of this engineering.

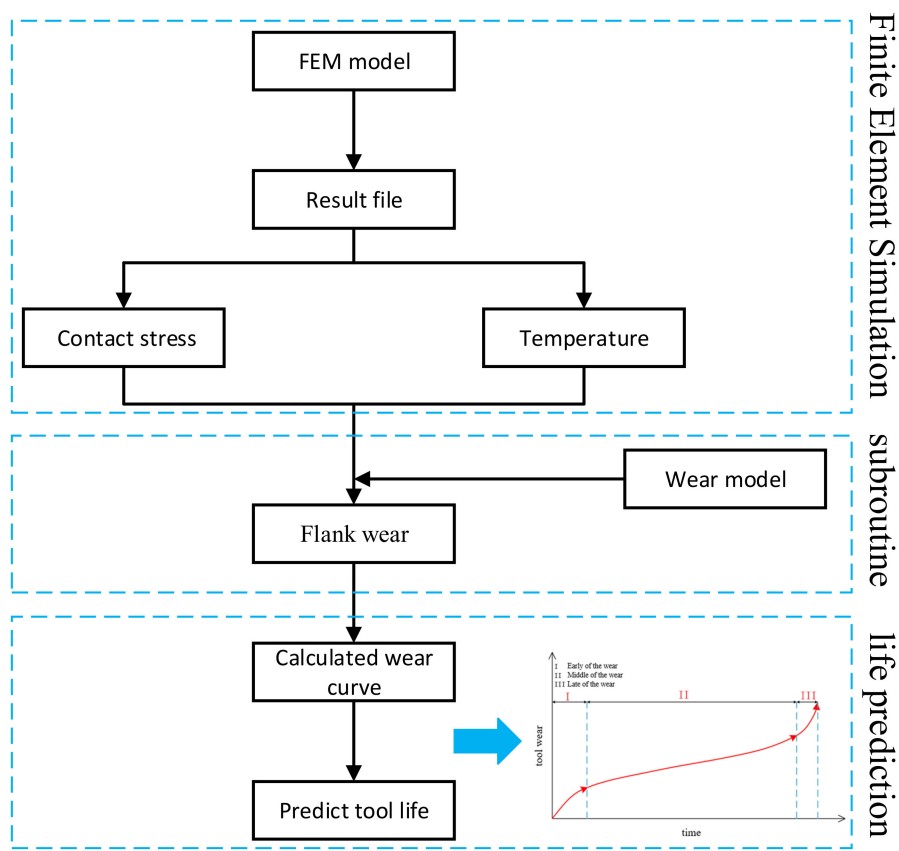

**Figure 1.** Flow chart of abrasion simulation.

## 2.2. Establishment of Milling Simulation Model

### 2.2.1. Ti6Al4V Constitutive Model

The [19] of Johnson–Cook constitutive model is adopted to describe the material flow stress of work piece, and such model expresses equivalent stress, equivalent plastic strain, equivalent plastic strain rate, and temperature, with the model expression formula shown in Figure 1:

$$\bar{\sigma} = \left[A + B(\bar{\varepsilon})^n\right]\left[1 + Cln\left(\frac{\dot{\bar{\varepsilon}}}{\dot{\bar{\varepsilon}}_0}\right)\right]\left[1 - \left(\frac{T - T_r}{T_m - T_r}\right)^m\right] \tag{1}$$

where $\bar{\sigma}, \bar{\varepsilon}, \dot{\bar{\varepsilon}}, \dot{\bar{\varepsilon}}_0$ represent the equivalent flow stress, equivalent plastic strain, equivalent plastic strain rate, and reference strain rate; *A*, *B*, *n*, *C* and *m*, as material constants, represent the yield strength, strain strengthening coefficient, strain hardening parameter, strain rate sensitivity coefficient, and thermal softening coefficient in quasi-static condition, respectively, which can be calculated through a tensile compression test of the materials; $T_r$ represents room temperature, at 25 °C in general; $T_m$ represents melting temperature of materials. The material parameter of Ti6Al4V adopted in this paper is gotten through the Hopkinson pressure lever experiment, with the specific parameters as shown in Table 1.

**Table 1.** Johnson–Cook constitutive parameter of Ti6Al4V [19].

| Material | *A/MPa* | *B/MPa* | *C* | *n* | *m* |
|----------|---------|---------|-----|-----|-----|
| T6Al4V | 543.75 | 1363.6 | 0.127 | 0.33 | 0.303 |

### 2.2.2. Friction Model

In the cutting process, the rack face and flank surface of the tool have friction with cuttings and work piece, respectively, the friction coefficient is related to the cutting temperature, cuttings sliding speed, and normal stress of the surface, and the friction size and distribution features influence the cutting temperature, cutting stress, tool abrasion, surface integrity, etc. In order to embed the compound abrasion model in the simulation model, the coefficient of local friction is needed at the contact part between tool and work piece. Currently, the friction model proposed by Wang, X [20] is adopted to describe the relationship between temperature and friction coefficient in the cutting process. The friction model is shown in Formula (2):

$$u(T) = 0.41 - \frac{0.103(T - 25)}{1000} \tag{2}$$

where the e u represents the friction coefficient and T represents the temperature (unit: °C). Formula (2) shows the friction coefficient decreases with the increase of temperature. Figure 2a shows the stress distribution curve on contact surface of cuttings in the temperature-related friction model, in which the normal stress changes constantly and continuously with different contact points of cuttings, reaches the maximum value at tool nose, and then gradually decreases and reaches 0 at the tool-cuttings separation position. Figure 2b [20] shows the approximation relation existing between the friction coefficient and friction temperature as shown in Formula (2), in which the friction coefficient can be deemed as the temperature-related function.

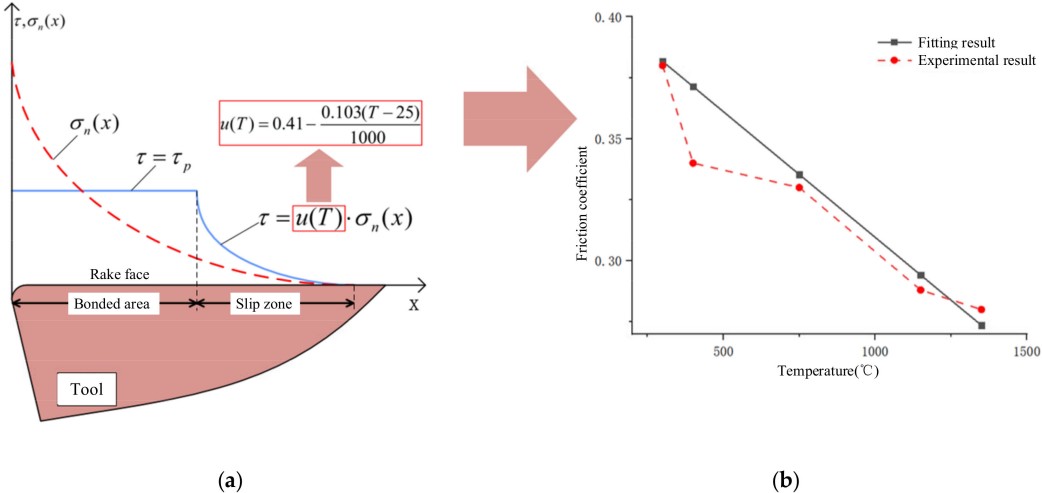

(**a**)                                      (**b**)

**Figure 2.** Temperature-related friction model (**a**) Stress distribution of tool-cuttings contact surface (**b**) Relationship between friction coefficient and temperature.

### 2.2.3. Damage Model

The Johnson–Cook damage model is adopted in this paper and can accurately reflect the failure mechanism of common metal materials. The model formula is shown in Formula (3) [19]:

$$\omega = \sum_{j=1}^{n}\left(\frac{\Delta\overline{\varepsilon^{pl}}}{\overline{\varepsilon_D^{pl}}}\right) \tag{3}$$

where $\Delta\overline{\varepsilon^{pl}}$ represents the equivalent plastic strain increments; $\overline{\varepsilon_D^{pl}}$ represents the failure strain.

If the materials in Formula (1) have failure behaviors, $\omega > 1\overline{\varepsilon_D^{pl}}$ the failure strain formula is shown in Formula (4) [19]:

$$\overline{\varepsilon_D^{pl}} = \left[d_1 + d_2\exp(d_3\frac{\sigma_p}{\sigma_{\min}})\right]\left[1+d_4\ln\left(\frac{\dot{\overline{\varepsilon}}}{\dot{\overline{\varepsilon_0}}}\right)\right]\left[1+d_5\left(\frac{T-T_{\text{room}}}{T_{\text{melt}}-T_{\text{room}}}\right)\right] \tag{4}$$

where $\dot{\overline{\varepsilon_0}}$ represents the reference strain rate; $\dot{\overline{\varepsilon}}$ represents the plastic strain rate; $d_1$, $d_2$, $d_3$, $d_4$, and $d_5$ represent the failure parameters of materials.

The failure parameter value of titanium alloy TC4 researched in this paper is shown in Table 2:

**Table 2.** Johnson–Cook failure parameter of TC4 [19].

| d1 | d2 | d3 | d4 | d5 |
|---|---|---|---|---|
| −0.09 | 0.25 | −0.5 | 0.014 | 3.87 |

### 2.2.4. Cuttings Separation Criterion

The metal cutting process is accompanied by the deformation and separation of materials, so the reasonable separation criterion of cuttings is needed to accurately reflect the mechanical property and physical property of work piece material. The cuttings separation criterion adopted in this paper

regards energy as cuttings separation standards, with the mathematical model shown in Formula (5) [19]:

$$\begin{cases} D = \frac{\int_0^u \bar{\sigma} d\overline{u^{pl}}}{G_f} \\ G_f = \frac{1-v^2}{E} K^2 \end{cases} \tag{5}$$

where $\overline{u^{pl}}$ represents the equivalent plastic displacement; $D$ represents the failure displacement; $G_f$ represents the fracture energy; $K$ represents the fracture toughness of materials, and the final failure displacement is calculated as 0.005 mm.

### 2.3. Grid Partition of Finite Element Model

Macroscopically, milling processing is an intermittent cutting process, and the action point of cutting force changes constantly. Microscopically, the milling can be simplified as orthogonal cutting or bevel cutting. In the finite element simulation, the cutting process is simulated mainly through the element failure occurred upon the contact between tool and work piece element.

In the plastic deformation containing thermal coupling simulation, the ABAQUS/Explicit module is usually adopted for analysis, and the dimension and type of grid are related to the precision of the simulation results. In the three-dimensional plane of pre-processing with work piece dimension of 60 mm × 40 mm × 40 mm, the work piece adopts CPE4RT grid and contains 800,000 regular tetrahedron element. The insulation lasts for about 15–20 h, and the established finite element model is shown in Figure 3. The tool adopts the overall end mill with a rake angle, clearance angle, and helical angle of 8°, 9°, and 55°, respectively, and the blunt diameter of cutting edge of 0.015 mm. The milling processing parameters are as follows: cutting depth: 6 mm, cutting width: 3 mm, feed speed: 400 mm/min and rotating speed of tool: 2000 r/min.

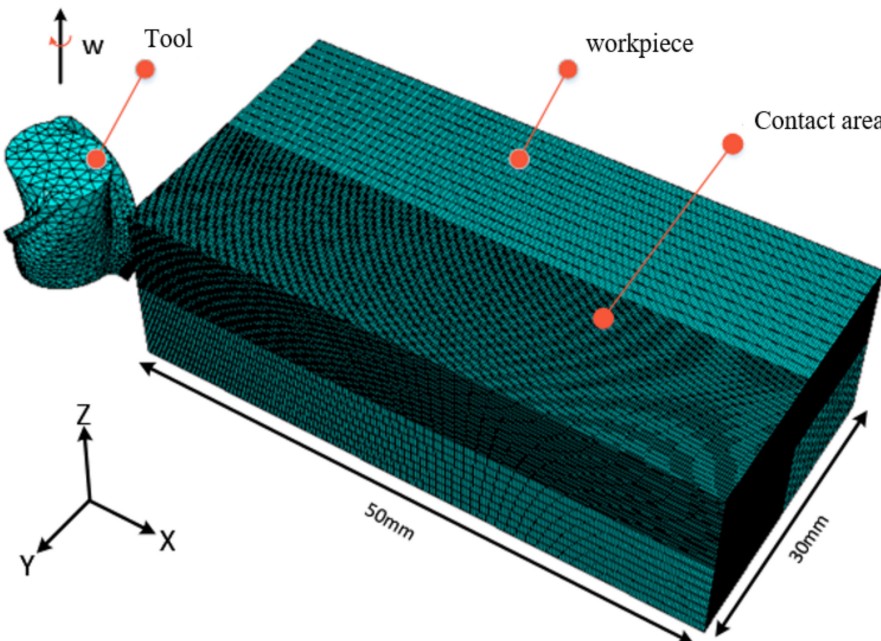

**Figure 3.** Finite element simulation model.

In the pre-processing, the tool adopts the wedge-shaped grid. The tool grid will be moderate in size because an over-sized grid would influence the value simulation precision and an under-sized grid would generate negative volume in update process of grid, as shown in Figure 4. If there is a larger displacement at the cutting-edge node, the nodes on surface would move to other elements which generate negative volume. The large grid would influence simulation precision. As such, the grid shall adopt the wedge-shaped element with side length of 0.1 mm after comprehensive consideration.

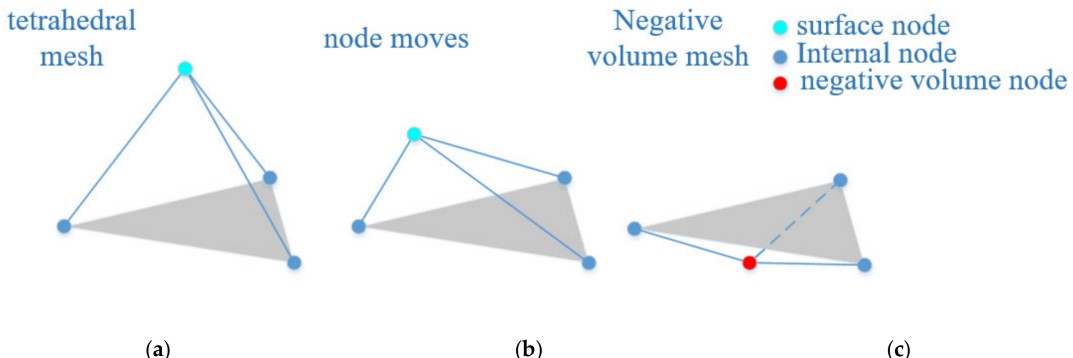

**Figure 4.** Schematic diagram of node displacement; (**a**) tetrahedron element (**b**) node displacement element (**c**) element of negative volume.

*2.4. Prediction of Tool Abrasion Course and Tool Service*

In order to predict tool abrasion course and tool service, the empirical formula of tool abrasion is hereby introduced, in which the finite element simulation results are put to work out the corresponding coefficient in formula. The metal cutting principle shows that there is complex exponential relation between abrasion loss *VB* of flank surface of tool and cutting parameter based on the confirmed machine tool characteristics and the geometrical parameter of tool, and the empirical formula of flank surface abrasion can be expressed in Formula (6):

$$VB(t) = kv^{a1} f_z{}^{a2} a_p{}^{a3} a_e{}^{a4} t^{a5} \qquad (6)$$

where *VB* represents the abrasion value of tool flank surface in simulation, and *k* represents abrasion coefficient of tool flank surface which is decided by work piece materials, tool materials, and tool structure and refers to the abrasion influence index of tool flank surface, including the rotating speed of main shaft, feed speed, axial cutting depth, radial cutting depth and cutting time. $a_1, a_2, a_3, a_4, a_5,$ $v, f_z, a_p, a_e, t$ take the logarithm on both sides in Formula (6) to form linear function, namely:

$$\lg VB = \lg k + a1\lg v^{a1} + a2\lg f_z + a3\lg a_p \\ + a4\lg a_e + a5\lg t \qquad (7)$$

Then $\begin{array}{l} y = \lg VB, a_0 = \lg k, x_1 = \lg v, x_2 = \lg f_z, \\ x_3 = \lg a_p, x_4 = \lg a_e, x_5 = \lg t \end{array}$ .

$$y = a_0 + a_1 x_1 + a_2 x_2 + a_3 x_3 + a_4 x_4 + a_5 x_5 \qquad (8)$$

The multiple linear regression equation is worked out as per Formula (8):

$$\left\{ \begin{array}{l} y_1 = A_0 + A_1 x_1 + A_2 x_2 + A_3 x_3 + A_4 x_4 + A_5 x_5 + \varepsilon_1 \\ y_2 = A_0 + A_1 x_6 + A_2 x_7 + A_3 x_8 + A_4 x_9 + A_5 x_{10} + \varepsilon_2 \\ \cdots\cdots\cdots \\ y_n = A_0 + A_1 x_{n+1} + A_2 x_{n+2} + A_3 x_{n+3} + A_4 x_{n+4} + A_5 x_{n+5} + \varepsilon_n \end{array} \right. \qquad (9)$$

where $\varepsilon$ is the random error of simulation.

Formula (9) adopts matrix form and is expressed as:

$$Y = AX + \varepsilon \tag{10}$$

where $Y = \begin{Bmatrix} y_1 \\ y_2 \\ \dots \\ y_n \end{Bmatrix}$, $A = \begin{Bmatrix} A_1 \\ A_2 \\ \dots \\ A_5 \end{Bmatrix}$ $X = \begin{Bmatrix} 1 & x_{11} & x_{12} & x_{13} & x_{14} & x_{15} \\ 1 & x_{21} & x_{22} & x_{23} & x_{24} & x_{25} \\ 1 & \dots & \dots & \dots & \dots & \dots \\ 1 & x_{n+1} & x_{n+2} & x_{n+3} & x_{n+4} & x_{n+5} \end{Bmatrix}$ $\varepsilon = \begin{Bmatrix} \varepsilon_1 \\ \varepsilon_2 \\ \dots \\ \varepsilon_n \end{Bmatrix}$.

The least square method is adopted to evaluate parameters. $A^*$ is evaluated with the least square method, with tropic equation as follows: $a_1, a_2, a_3, a_4, a_5$ $A_1, A_2, A_3, A_4, A_5$

$$\overline{y} = a_0 + a_1 x_1 + a_2 x_2 + a_3 x_3 + a_4 x_4 + a_5 x_5 \tag{11}$$

In the formula, $\overline{y}$ represents statistical variable and $a_0, a_1, a_2, a_3, a_4, a_5$ represents regression coefficient. Finally, the coefficient of the regression model is solved through MATLAB programming and the time-varying curve of tool abrasion is shown in Figure 5.

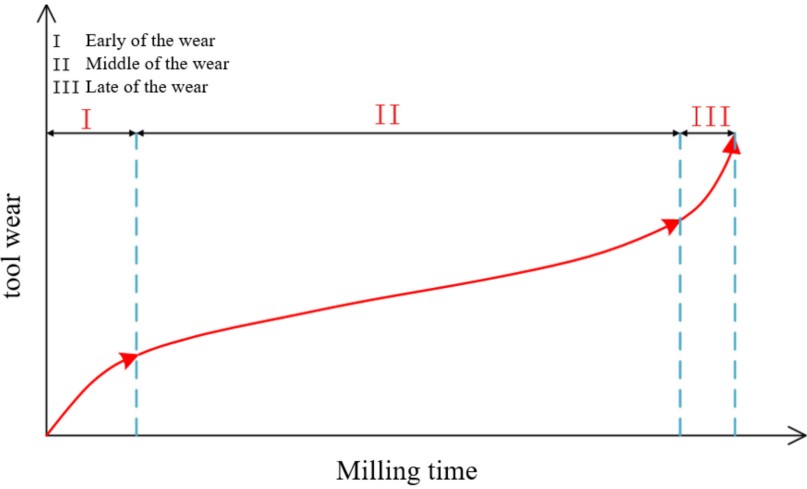

**Figure 5.** Time-varying curve of tool abrasion.

## 3. Expression of Different Types of Tool Abrasion Loss

The abrasion formula in subprogram is the core of the whole simulation program. Tool abrasion is a complex process due to coupling of several kinds of abrasion mechanism, and different abrasion mechanisms are mutually influenced. In the milling process, the tool mainly includes the following failure modes: tool abrasion, breakage, tipping, blade fracture, etc. The relationship between tool abrasion mechanism and temperature is shown in Figure 6 [21], which shows a close relationship between them. If the temperature is low, the surface of tool mainly involves grains abrasion and adhesion abrasion. If the cutting temperature is high, the surface of tool mainly involves diffusion abrasion and oxidation abrasion.

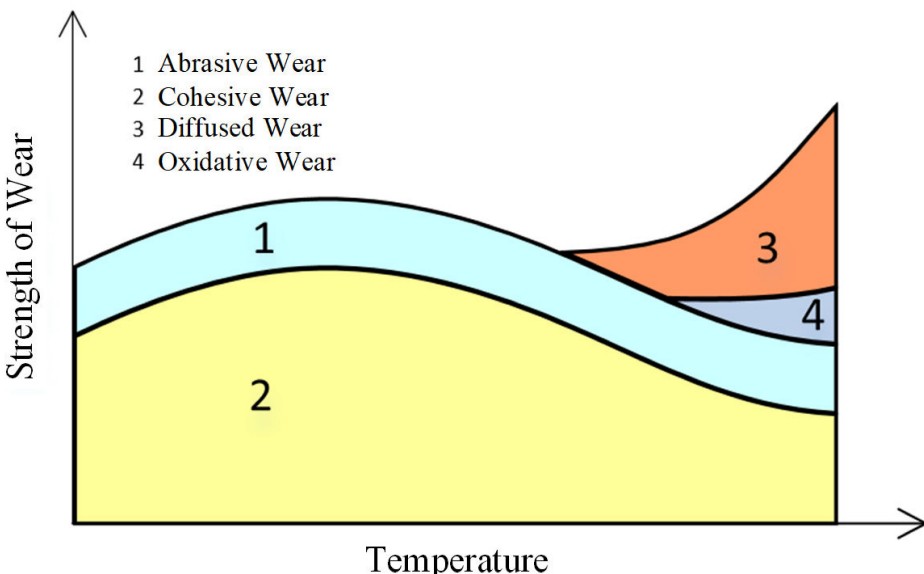

**Figure 6.** Relationship between tool abrasion mechanism and temperature.

### 3.1. Adhesion Abrasion (AD)

In the processing process of titanium alloy, tool abrasion is mainly divided into adhesion abrasion, grain abrasion, and diffusion abrasion. Many scholars certify that, in high-speed cutting condition, the adhesion abrasion widely applied in actual scene for carbide cutter adopts the model proposed by Usui et al. [22], as shown in Formula (12).

$$\frac{W_n}{dt} = A_w \cdot \sigma_n \cdot v_c \exp(-B_w/(273+T)) \tag{12}$$

where $W_n$ represents the adhesion abrasion loss, mm/min; $A_w, B_w$ represents the abrasion constant obtained through experiment, which mainly depends on the tool materials and work piece materials. $A_w, B_w$ indicates the titanium alloy cut with carbide cutter, its abrasion rate can be calculated through cutting experiment and its constant is calculated based on Formula (1). $\sigma_n$ represents the cutting stress on surface of tool, with unit of Mpa. $v_c$ represents the sliding velocity of work piece materials, with unit of mm/min. $T$ represents the cutting temperature, with unit of °C. In milling of titanium alloy TC4, $A_w$ and $B_w$ are the experience value in general, in which the $A_w$ is $7.8e^{-9}$ [20] and $B_w$ is 2500 [20]. Thus, the mathematical expression of final adhesion abrasion simulation is shown in Formula (13):

$$\frac{W_n}{dt} = 7.8 \times 10^{-9} \cdot \sigma_n \cdot v_c \exp(-2500/(273+T)) \tag{13}$$

### 3.2. Grains Abrasion (GA)

In the cutting process, there are micro hard particles on contact surface between tool and work piece. With the relative movement between tool and cuttings, the groove is left on the surface of tool, namely tool has grains abrasion which is mainly related to the grains shape, hardness and distribution condition. In that case, Rainowicz et al. [23] researched grains abrasion and put forward the grain abrasion formula, as shown in Formula (14):

$$\frac{W_a}{dt} = G \cdot v \tag{14}$$

where $W_a$ represents grain abrasion, $G$ represents grain abrasion constant, with unit of mm; V represents the relevant sliding velocity, with unit of mm/s. In milling of titanium alloy TC4, G [24] is $2.37e^{-11}$, thus, the final expression formula of grains abrasion is shown in Formula (15):

$$\frac{W_a}{dt} = 2.37 \times 10^{-11} v \tag{15}$$

### 3.3. Diffusion Abrasion (DA)

In the milling and processing process, the tool has low feed and a high rotating speed in general. Under this condition, there is a concentration difference and temperature difference in the elements in both tool and work piece, the elements in each part rapidly diffuse, and the diffusion phenomenon concentrates on the contact area between abrasion belt of tool face and processing surface, as shown in Figure 7.

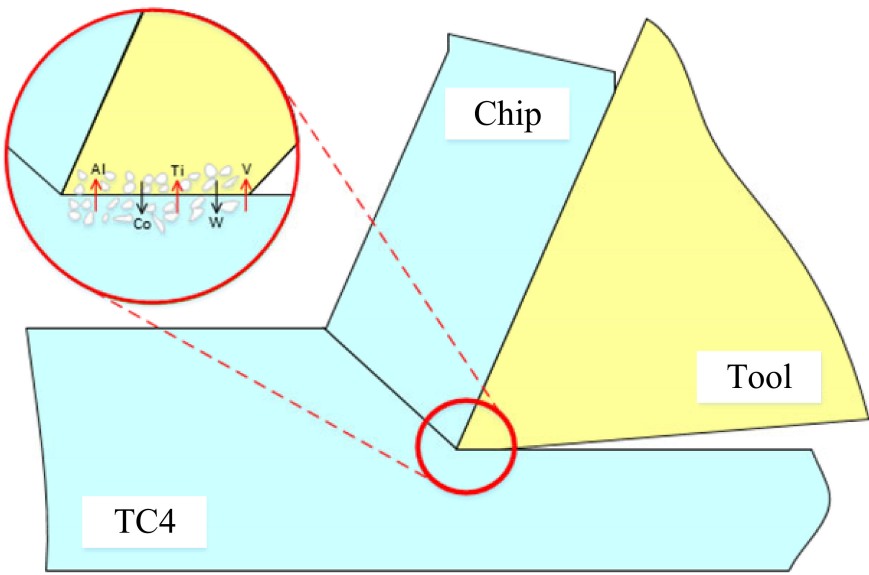

**Figure 7.** Schematic diagram of diffusion abrasion of the milling cutter flank surface.

For titanium alloy TC4, Sun, Y [25] researched the carbide cutter abrasion, analyzed the influence of temperature on tool abrasion mechanism, and established the tool abrasion model about temperature, with the diffusion abrasion formula of flank surface shown in Formula (16):

$$\frac{W_r}{dt} = \frac{2C_0}{\rho} \left( \frac{vD_0}{x\pi} \right)^{\frac{1}{2}} e^{-Q/2R_Q(273+T)} \tag{16}$$

where $W_a$ represents diffusion abrasion, $C_0$ represents the concentration of diffusion materials, $D_0$ represents equation coefficient, $Q$ represents activation energy, $\rho$ represents tool density, $v$ represents relevant sliding velocity between flank surface and processed surface, $x$ represents the distance between any point in tool-cuttings contact area and the cutting edge, RQ represents gas constant, and $T$ represents temperature, with unit of °C. The above physical constants can be obtained based on relevant literature [26], as follows: $C_0 = 0.0253$ mole/mm$^3$, $D_0 = 1.9$ mm$^2$/s, $Q = 114.4$ KJ/mole, $R_Q = 8.315e - 3$ KJ/mole/K and $\rho = 14.9e3$ kg/m$^3$. Other data can be obtained through experiment or simulation, with the final expression formula shown in Formula (17):

$$\frac{W_r}{dt} = 3.3 \times \left( \frac{0.61v}{x} \right)^{\frac{1}{2}} e^{-6861/T} \tag{17}$$

Figure 6 shows that there is mainly the grains abrasion and diffusion abrasion for interaction on the surface of tool when temperature is low, and the oxidation abrasion and diffusion abrasion occur when temperature is high. Based on that, Li, Y et al. [26], after experiment research, found that the element diffusion happens to cemented carbide and titanium alloy on the combination boundary when the cutting temperature reaches 600 °C. Molinari et al. [27] researched and found the grains abrasion can be neglected when the cutting temperature exceeds 800 °C. Based on the research of the above scholars, abrasion model can be divided into three stages based on temperature, as shown in Formula (18):

$$
\left\{
\begin{array}{l}
\frac{W}{dt} = \frac{W_n}{dt} + \frac{W_a}{dt} = 7.8 \times 10^{-9} \cdot \sigma_n \cdot v_c \exp(-2500/(273+T)) \\
\qquad\qquad + 2.37 \times 10^{-11} v \qquad (T < 600) \\
\frac{W}{dt} = \frac{W_n}{dt} + \frac{W_a}{dt} + \frac{W_r}{dt} = 7.8 \times 10^{-9} \cdot \sigma_n \cdot v_c \exp(-2500/(273+T)) \\
\qquad + 2.37 \times 10^{-11} v + 3.3 \times \left(\frac{0.61v}{x}\right)^{\frac{1}{2}} e^{-6861/T} \quad (600 < T < 800) \\
\frac{W}{dt} = \frac{W_a}{dt} + \frac{W_r}{dt} = 2.37 \times 10^{-11} v + 3.3 \times \left(\frac{0.61v}{x}\right)^{\frac{1}{2}} e^{-6861/T} \quad (T > 800)
\end{array}
\right\}. \tag{18}
$$

In order to simulate the occurrence of diffusion abrasion of tool when temperature is over 600 °C and the disappearing of diffusion abrasion when the temperature is over 800 °C, the temperature must be controlled. The grains-adhesion abrasion is adopted in the area with temperature less than 600 °C, grains-diffusion abrasion is adopted in the area with temperature between 600 °C–800 °C, and adhesion-diffusion abrasion is adopted in the area with temperature over 800 °C. The schematic diagram of temperature control is shown in Figure 8.

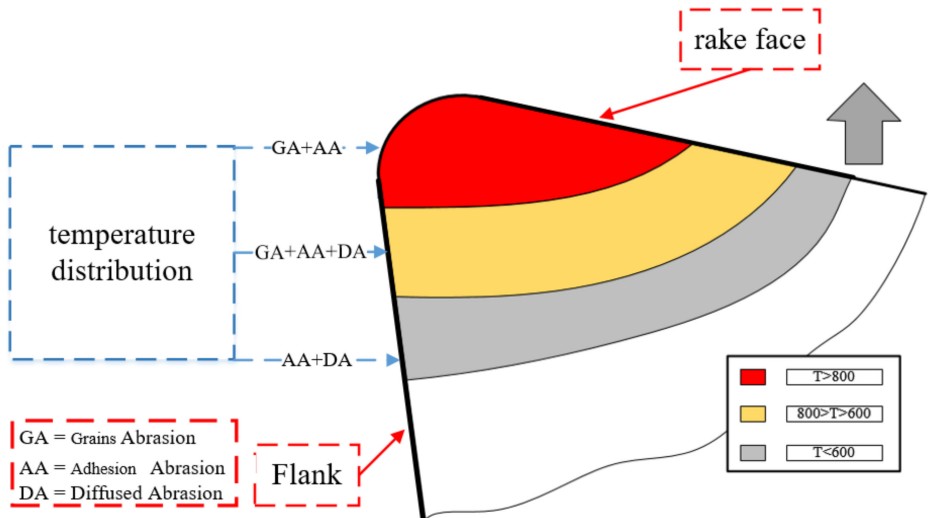

**Figure 8.** Schematic diagram of tool abrasion type at different temperatures.

## 4. Experimental Verification and Simulation Results

### 4.1. Milling Experiment Design

The processing experiment of milling is shown in Figure 9 and is conducted on the VDL-100E triaxial numerically controlled machine tool produced by Dalian Machine Tool Group. The designed experiment works out the same parameters as simulation model, namely, the cutting depth, milling width, feed speed, and rotating speed of main shaft are 6 mm, 3 mm, 400 mm/min, and 2000 r/min, respectively. Since the influence of cutting fluid on tool abrasion is not considered in the model, the dry cutting is adopted in this experiment. The test adopts the cylindrical spiral blade end mill as the tool, with the number of teeth, diameter, rake angle, clearance angle, and helical angle being 2, 10 mm, 8°, 9°, and 55°, respectively; The test work piece adopts titanium alloy TC4, with a dimension of 50

× 200 × 300 mm. KEYENCE VHX- super-depth-of-field microscope is adopted to measure the tool abrasion loss of flank surface of the milling cutter.

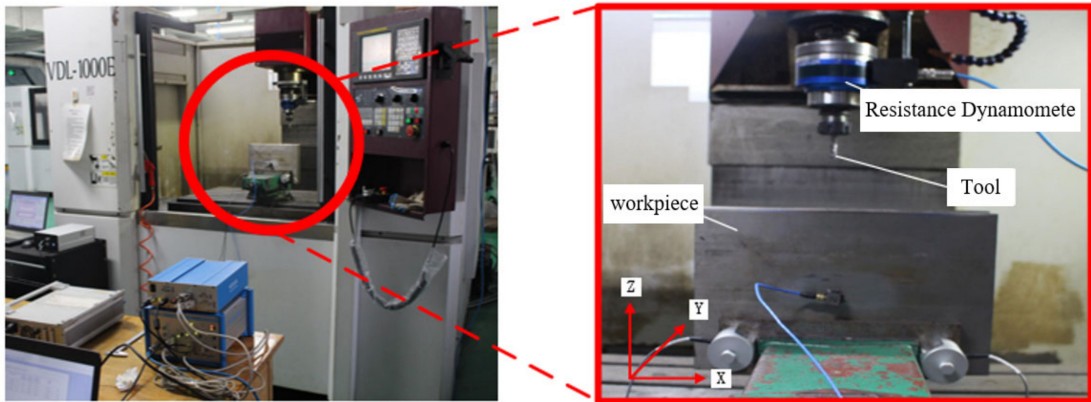

**Figure 9.** Milling processing test site.

## 4.2. Milling Experimental Results

In the experiment process, the abrasion characterization VB of tool flank surface is measured every eight minutes as per the experiment parameters, and the experiment measurement results are shown in Figure 10. The figure shows that the tool abrasion mainly occurs on the flank surface of the tool, and the abrasion depth increases with the increase of time. When tool abrasion reaches 113 um, the quality of work piece surface decreases, which does not meet the processing requirements. Hence, the time when tool abrasion reaches 113 um is deemed as the service life of the tool.

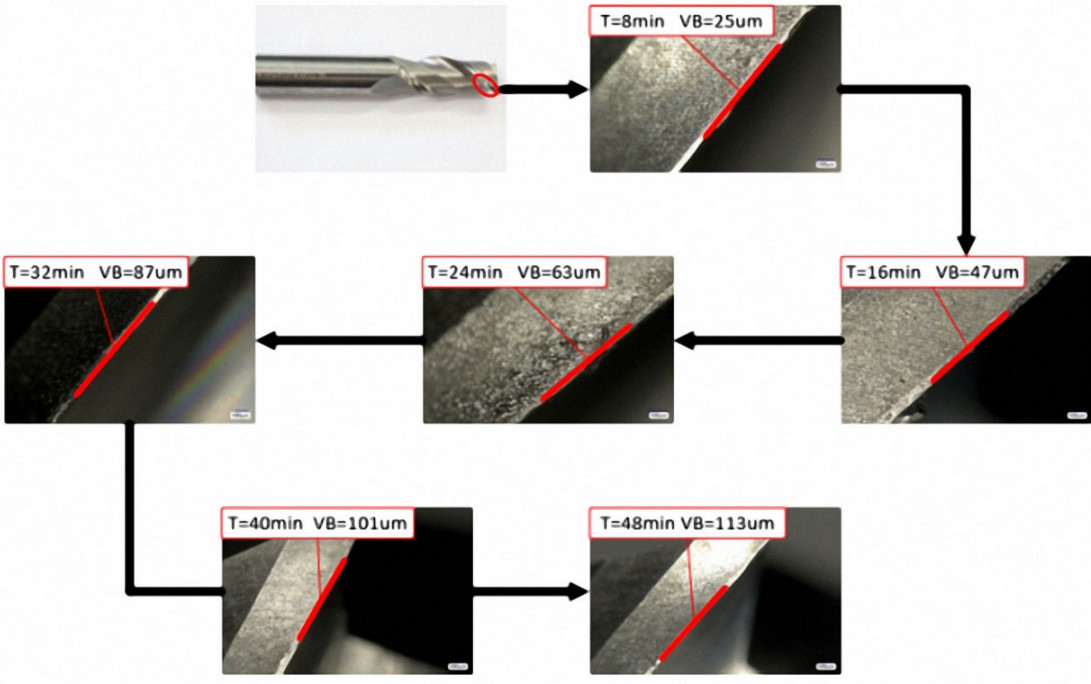

**Figure 10.** Experimental measurement results.

*4.3. Discussion of Finite Element Simulation Results*

4.3.1. Analysis of Temperature Field Results of Tool

As shown in Figure 11, Avg 75% in the picture means is default averaging threshold. When the first cutting edge of tool enters work piece, the temperature distribution results at the cutting edge of the tool show that the temperature does not rise sharply due to its hysteresis.

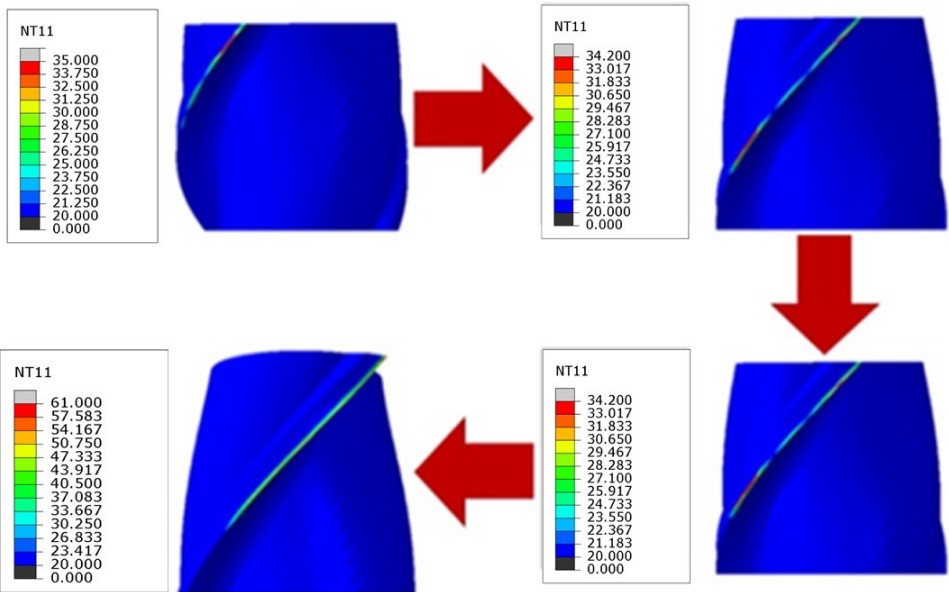

**Figure 11.** Simulation cloud diagram for cutting edge into workpiece.

When the cutting is stable, as shown in Figure 12, the temperature of the tool cutting edge is about 500 °C, but the temperature of the tool nose is higher than that of cutting edge and reaches about 1000 °C, mainly because, in the continuous cutting process, there is an extrusion-type cutting between tool nose and work piece at first; such a cutting method increases the friction effect between tool and work piece, then the heat concentrates on the tool nose and cannot disperse easily in a short amount of time. Meanwhile, in other areas, temperature mainly shows the zonal distribution along the cutting edge, and temperature gradually decreases along the tool nose toward the shaft center direction of tool. Thus, the tool nose will not have severe abrasion in the cutting process.

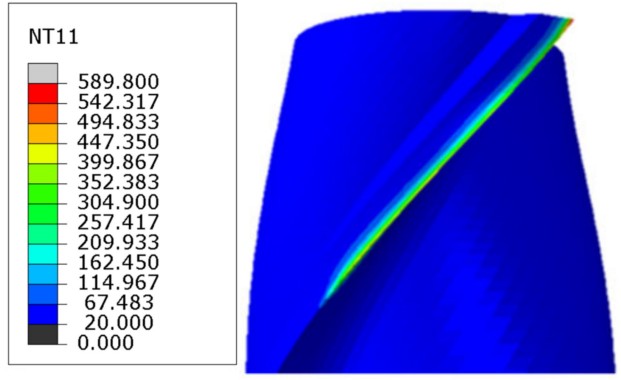

**Figure 12.** Temperature of the cutting edge after stabilization.

4.3.2. Analysis of Tool Abrasion Results

The program is put in the simulation model through user subprogram to calculate the tool abrasion loss, as show in Figure 13 which shows that the tool abrasion mainly occurs at the cutting edge with zonal distribution. There is uneven abrasion at the cutting edge, mainly due to larger contact stress at cutting edge in cutting process and accumulative abrasion with the increase of the cutting time.

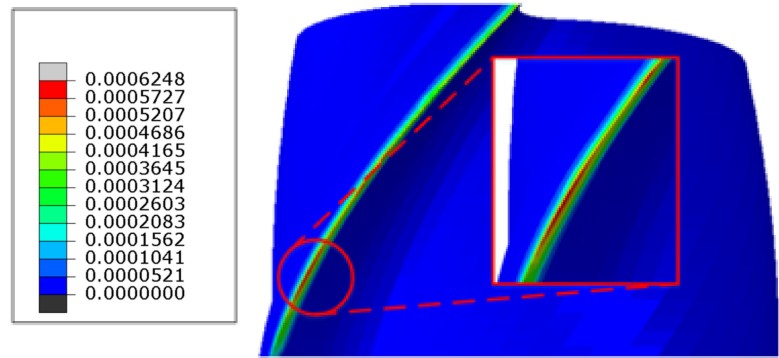

**Figure 13.** Cloud diagram of tool abrasion loss (unit: mm/10 s).

Based on the above method, the abrasion at different times is put in an empirical formula to calculate constant, with calculation results shown in Table 3. The simulation results and experiment comparison of abrasion characterization VB for the milling cutter flank surface are shown in Figure 14.

**Table 3.** Parameters of the empirical formula.

| $k$ | $a_1$ | $a_2$ | $a_3$ | $a_4$ | $a_5$ |
| --- | --- | --- | --- | --- | --- |
| 0.7573 | −7.277 | 3.2676 | 9.6475 | 8.3551 | 0.9567 |

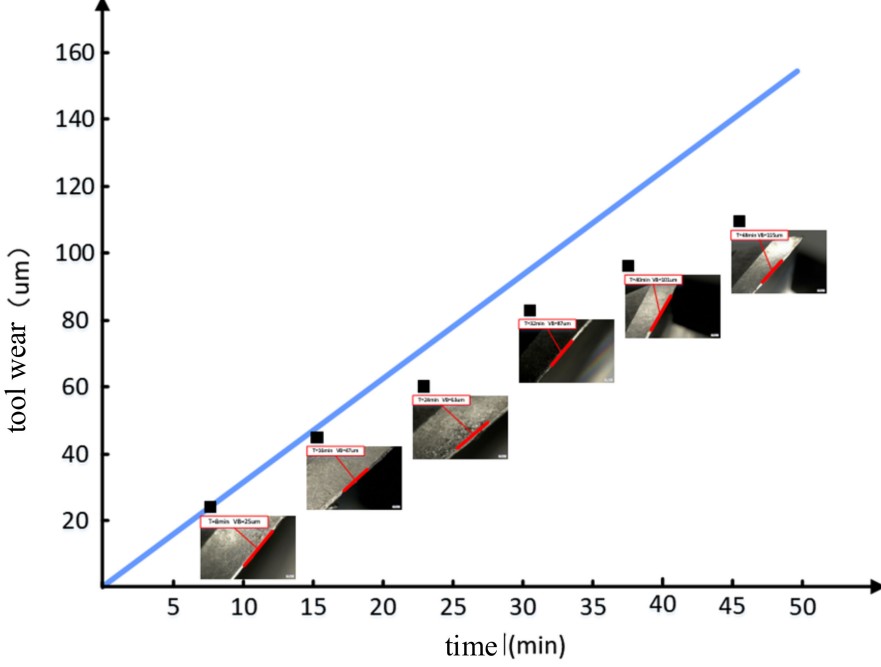

**Figure 14.** Comparison of abrasion VB between simulation and experiment for milling cutter flank surface.

The final expression formula is:

$$VB = 0.7573v^{-7.2776} f_z{}^{3.2676} a_p{}^{9.6475} a_e{}^{8.3551} t^{0.9567} \tag{19}$$

Figure 14 shows that the experiment results of end mill flank surface abrasion VB keep consistent with the change trend, which verifies the feasibility of such method. Figure 14 shows that the predicted value is larger than the actual measured value, mainly because the predicted simulation abrasion mainly refers to preliminary abrasion, which is relatively fierce, without stable periods of medium-term abrasion. The error of this method is within 30%, mainly due to the following factors:

(1) Error caused by grid dimension: The grid dimension decides the simulation precision. Oversized grid is difficult to guarantee the simulation precision and undersized grid causes an overly long simulation time. Hence, the appropriate grid dimension shall be selected after comprehensive consideration.

(2) Milling vibration factors are not considered in the established simulation process of finite element simulation model.

(3) The tool abrasion caused due to tipping is not considered in simulation process, and the tool Mises press is mainly observed to find tool tripping in the finite element simulation.

(4) In the milling process of cemented carbide end mill, there is oxidation abrasion on its flank surface, but the oxidation abrasion is not considered in the abrasion prediction model.

## 5. Conclusions

The simulation model of tool abrasion and the prediction model of tool service life based on finite element simulation method are established on account of the cutting processing process of Ti6Al4V to work out the distribution condition of the tool nose temperature and contact stress and to accurately predict the abrasion condition of tool flank surface. The research results provide reasonable technical support for monitoring tool abrasion status, predicting tool service life and optimizing tool structure. The conclusion is as follows:

(1) On account of the change of tool abrasion type at different temperatures, the carbide cutter abrasion model considering temperature effect is constructed to avoid the limitation of single model and improve the prediction precision of the tool abrasion;

(2) Combined the simulation results with the empirical formula, the tool abrasion course function can be calculated, which saves lots of simulation time and realizes the rapid prediction of the tool's service life;

(3) The test about on the service life of tool is carried out, and the simulation results and experiment measurement results are compared and analyzed. The simulation results can better simulate the change rules of tool abrasion in cutting process, and prediction error is within 30%, which can predict the service life of the tool to some extent.

**Author Contributions:** Z.L., C.Y.: contributed to the conception of the study; Z.L., X.L. (Xiaochen Li): performed the experiment; Z.L., C.Y., S.Y.L., X.L. (Xianli Liu): contributed significantly to analysis and manuscript preparation; C.Y., S.Y.L., L.W., X.L. (Xianli Liu): helped perform the analysis with constructive discussions. All authors have read and agreed to the published version of the manuscript.

**Funding:** This research was funded by Natural Science Outstanding Youth Fund of Heilongjiang Province (Grant Number YQ2019E029; Outstanding Youth Project of Science and Technology Talents, grant number LGYC2018JQ015; National Natural Science Foundation of China, grant number: 5171001055, National key research and development programs, grant number: 2019YFB1704800.

**Acknowledgments:** Thanks are due to Chen Zhitao for assistance with the experiments.

**Conflicts of Interest:** The authors declare no conflict of interest.

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
