# Peer review of "Research on Tool Wear Based on 3D FEM Simulation for Milling Process"

_jmmp, doi:10.3390/jmmp4040121_

Round 1

Reviewer 1 Report

This paper presented an interesting research on tool wear condition and service life prediction, this topic is quite important and the research has potential application for industry. Different types of tool failure are predicted by finite element simulation, and according to the empirical formula, the tool failure process is established to predict the tool service life. In this study, multiple tool failure modes are considered to reflect the tool condition more accurately, good results have been achieved. Therefore, this study is a very effective method for tool life assessment. The paper is well organized and technical sound. Some detailed comments on this paper are as follows:

(1) Data-driven method has been widely applied for tool wear prediction, it is suggested to include some literatures on this aspect.

(2) How many simulation data sets are required for fit the empirical formula? Is it helpful if some experiment data are added for fitting the empirical formula?

(3) In the prediction of tool abrasion course and tool service section, it’s more appropriate to change VB to VB(t) in formula 6.

(4) In the milling experimental results section, how is the actual tool wear obtained and what is the accuracy?

(5) In the analysis of temperature field results of the tool section, why is the tip temperature high but wouldn’t have severe abrasion in cutting process? Please give some explanation.

(6) There is only one cutting condition is verified, what is the feasiblity range of the cutting conditions, for example, tool abrasion type under different cutting parameters.

Author Response

The following is a point-to-point response to the two reviewers’ comments. 

Q: (1) Data-driven method has been widely applied for tool wear prediction, it is suggested to include some literatures on this aspect.

Answer: The reviewer and editor’s suggestions have been adopted and the conclusions have been changed in Page 2 line 91.

Q: (2) How many simulation data sets are required for fit the empirical formula? Is it helpful if some experiment data are added for fitting the empirical formula?

Answer: 200 simulation data can be obtained through finite element simulation,and the more the simulation data we obtain , the more accurate the fitting result will be.

Q: (3) In the prediction of tool abrasion course and tool service section, it’s more appropriate to change VB to VB(t) in formula 6.

Answer: The reviewer and editor’s suggestions have been adopted and the conclusions have been changed in Page 7 line 215.

Q: (4) In the milling experimental results section, how is the actual tool wear obtained and what is the accuracy?

Answer: The tool flank wear is mainly to measure on the KEYENCE VHX- super-depth-of-field microscope with an accuracy of about 95 percent.

Q: (5) In the analysis of temperature field results of the tool section, why is the tip temperature high but wouldn’t have severe abrasion in cutting process? Please give some explanation.

Answer: The tip temperature is the highest in the whole process.But in the calculation process, both of the temperature and the stress will have an important effect on tool wear.Because of the contact stress on the flank surface of the tool is greater, the wear on the flank is more severe.

Q: (6) There is only one cutting condition is verified, what is the feasiblity range of the cutting conditions, for example, tool abrasion type under different cutting parameters.

Answer: Due to the limitation of wear formula, this formula is only applicable to end milling. If other machining methods are used to predict tool wear, some parameters in the wear model need to be modified.

Reviewer 2 Report

The presented in the manuscript numerical model of tool abrasion and the regression model of tool service life extends the existing area of knowledge about the mechanism of tool wear and can be used to predict tool wear and optimization of the process. The topic is quite interesting but the manuscript requires minor revisions before it can be accepted for publication:
1. Please increase the resolution of Figures 1, 10, and 14 – the legend and descriptions of the axis are fuzzy. Please zoom pictures in figure 10.
2. Please give more information about the experiment setup – please specify the assumed confidence level of the flank surface abrasion VB regression model.
3. Please correct in line 313: add Fig. before 10
After going through the paper, my general opinion are that the article is quite interesting and presents an important topic.

Author Response

The following is a point-to-point response to the two reviewers’ comments. 

Q1.Please increase the resolution of Figures 1, 10, and 14 – the legend and descriptions of the axis are fuzzy. Please zoom pictures in figure 10.

Answer: The reviewer and editor’s suggestions have been adopted and the conclusions have been changed in Page 3 line 122;. Page 12 line 327; Page 14 line 358.

Q2.Please give more information about the experiment setup – please specify the assumed confidence level of the flank surface abrasion VB regression model.

Answer:The designed experiment works out the same parameters as simulation model, namely the cutting depth, milling width, feed speed and rotating speed of main shaft are 6mm, 3mm, 400mm/min and 2,000r/min, respectively.

 The test adopts the cylindrical spiral blade end mill as tool, with the number of teeth, diameter, rake angle, clearance angle and helical angle of 2, 10mm, 8°, 9° and 55°, respectively; KEYENCE VHX- super-depth-of-field microscope is adopted to measure the tool abrasion loss of flank surface of milling cutter.

Figure 1. KEYENCE VHX- super-depth-of-field microscope.

Figure 2.The same end mills involved in the experimen

The above experiment was repeated three times while the machine tool, end millings, workpiece and cutting parameters are the same.KEYENCE VHX- super-depth-of-field microscope is adopted to measure the tool abrasion loss of flank surface of milling cutter. According to international standards, the posotion at 1/2 cutting depth is selected to measure the average wear on flank ,which the measured value is the average of the three measurements.According to the original experimental data, The confidence level of the wear on the flank changing with time is 0.807, which proved that the experimental data is reasonable.

Q3.Please correct in line 313: add Fig. before 10

Answer: The reviewer and editor’s suggestions have been adopted and the conclusions have been changed in Page 12 line 323.